# Learning Causal Graphs with Small Interventions

**Karthikeyan Shanmugam**[1]**, Murat Kocaoglu**[2]**, Alexandros G. Dimakis**[3]**, Sriram Vishwanath**[4]
Department of Electrical and Computer Engineering
The University of Texas at Austin, USA
[1]`karthiksh@utexas.edu,`[2]`mkocaoglu@utexas.edu,`
[3]`dimakis@austin.utexas.edu,`[4]`sriram@ece.utexas.edu`

## Abstract

We consider the problem of learning causal networks with interventions, when each intervention is limited in size under Pearl's Structural Equation Model with independent errors (SEM-IE). The objective is to minimize the number of experiments to discover the causal directions of all the edges in a causal graph. Previous work has focused on the use of separating systems for complete graphs for this task. We prove that any deterministic adaptive algorithm needs to be a separating system in order to learn complete graphs in the worst case. In addition, we present a novel separating system construction, whose size is close to optimal and is arguably simpler than previous work in combinatorics. We also develop a novel information theoretic lower bound on the number of interventions that applies in full generality, including for randomized adaptive learning algorithms.

For general chordal graphs, we derive worst case lower bounds on the number of interventions. Building on observations about induced trees, we give a new deterministic adaptive algorithm to learn directions on any chordal skeleton completely. In the worst case, our achievable scheme is an $\alpha$-approximation algorithm where $\alpha$ is the independence number of the graph. We also show that there exist graph classes for which the sufficient number of experiments is close to the lower bound. In the other extreme, there are graph classes for which the required number of experiments is multiplicatively $\alpha$ away from our lower bound.

In simulations, our algorithm almost always performs very close to the lower bound, while the approach based on separating systems for complete graphs is significantly worse for random chordal graphs.

## 1   Introduction

Causality is a fundamental concept in sciences and philosophy. The mathematical formulation of a *theory of causality* in a probabilistic sense has received significant attention recently (e.g. [1–5]). A formulation advocated by Pearl considers the *structural equation models*: In this framework, $X$ is a cause of $Y$, if $Y$ can be written as $f(X, E)$, for some deterministic function $f$ and some latent random variable $E$. Given two causally related variables $X$ and $Y$, it is not possible to infer whether $X$ *causes* $Y$ or $Y$ *causes* $X$ from random samples, unless certain assumptions are made on the distribution of $E$ and/or on $f$ [6, 7]. For more than two random variables, *directed acyclic graphs* (DAGs) are the most common tool used for representing causal relations. For a given DAG $D = (V, E)$, the directed edge $(X, Y) \in E$ shows that $X$ is a cause of $Y$.

If we make no assumptions on the data generating process, the standard way of inferring the causal directions is by performing experiments, the so-called *interventions*. An intervention requires modifying the process that generates the random variables: The experimenter has to enforce values on the random variables. This process is different than conditioning as explained in detail in [1].

The natural problem to consider is therefore minimizing the *number* of interventions required to learn a causal DAG. Hauser et al. [2] developed an efficient algorithm that minimizes this number in the worst case. The algorithm is based on optimal coloring of chordal graphs and requires at most $\log \chi$ interventions to learn any causal graph where $\chi$ is the chromatic number of the chordal skeleton.

However, one important open problem appears when one also considers the *size* of the used interventions: Each intervention is an experiment where the scientist must force a set of variables to take random values. Unfortunately, the interventions obtained in [2] can involve up to $n/2$ variables. The simultaneous enforcing of many variables can be quite challenging in many applications: for example in biology, some variables may not be enforceable at all or may require complicated genomic interventions for each parameter.

In this paper, we consider the problem of learning a causal graph when intervention sizes are bounded by some parameter $k$. The first work we are aware of for this problem is by Eberhardt et al. [3], where he provided an achievable scheme. Furthermore [8] shows that the set of interventions to fully identify a causal DAG must satisfy a specific set of combinatorial conditions called *a separating system*[1], when the intervention size is not constrained or is 1. In [4], with the assumption that the same holds true for any intervention size, Hyttinen et al. draw connections between causality and known separating system constructions. One open problem is: If the learning algorithm is *adaptive* after each intervention, is a separating system still needed or can one do better? It was believed that adaptivity does not help in the worst case [8] and that one still needs a separating system.

**Our Contributions:** We obtain several novel results for learning causal graphs with interventions bounded by size $k$. The problem can be separated for the special case where the underlying undirected graph (the skeleton) is the complete graph and the more general case where the underlying undirected graph is chordal.

1. For complete graph skeletons, we show that any adaptive deterministic algorithm needs a $(n, k)$ separating system. This implies that lower bounds for separating systems also hold for adaptive algorithms and resolves the previously mentioned open problem.
2. We present a novel combinatorial construction of a separating system that is close to the previous lower bound. This simple construction may be of more general interest in combinatorics.
3. Recently [5] showed that *randomized* adaptive algorithms need only $\log \log n$ interventions with high probability for the unbounded case. We extend this result and show that $O\left(\frac{n}{k} \log \log k\right)$ interventions of size bounded by $k$ suffice with high probability.
4. We present a more general information theoretic lower bound of $\frac{n}{2k}$ to capture the performance of such randomized algorithms.
5. We extend the lower bound for adaptive algorithms for general chordal graphs. We show that over all orientations, the number of experiments from a $(\chi(G), k)$ separating system is needed where $\chi(G)$ is the chromatic number of the skeleton graph.
6. We show two extremal classes of graphs. For one of them, the interventions through $(\chi, k)$ separating system is sufficient. For the other class, we need $\frac{\alpha(\chi-1)}{2k} \approx \frac{n}{2k}$ experiments in the worst case.
7. We exploit the structural properties of chordal graphs to design a new deterministic adaptive algorithm that uses the idea of separating systems together with adaptability to Meek rules. We simulate our new algorithm and empirically observe that it performs quite close to the $(\chi, k)$ separating system. Our algorithm requires much fewer interventions compared to $(n, k)$ separating systems.

## 2 Background and Terminology

### 2.1 Essential graphs

A *causal DAG* $D = (V, E)$ is a directed acyclic graph where $V = \{x_1, x_2 \dots x_n\}$ is a set of random variables and $(x, y) \in E$ is a directed edge if and only if $x$ is a direct *cause* of $y$. We adopt Pearl's *structural equation model with independent errors* (SEM-IE) in this work (see [1] for more details).

Variables in $S \subseteq V$ *cause* $x_i$, if $x_i = f(\{x_j\}_{j \in S}, e_y)$ where $e_y$ is a random variable independent of all other variables.

The causal relations of $D$ imply a set of conditional independence (CI) relations between the variables. A conditional independence relation is of the following form: Given $Z$, the set $X$ and the set $Y$ are conditionally independent for some disjoint subsets of variables $X, Y, Z$. Due to this, causal DAGs are also called *causal Bayesian networks*. A set $V$ of variables is Bayesian with respect to a DAG $D$ if the joint probability distribution of $V$ can be factorized as a product of marginals of every variable conditioned on its parents.

All the CI relations that are learned statistically through observations can also be inferred from the Bayesian network using a graphical criterion called the *d-separation* [9] assuming that the distribution is faithful to the graph [2]. Two causal DAGs are said to be *Markov equivalent* if they encode the same set of CIs. Two causal DAGs are Markov equivalent if and only if they have the same skeleton[3] and the same immoralities[4]. The class of causal DAGs that encode the same set of CIs is called the *Markov equivalence class*. We denote the Markov equivalence class of a DAG $D$ by $[D]$. The graph union[5] of all DAGs in $[D]$ is called the *essential graph* of $D$. It is denoted $\mathcal{E}(D)$. $\mathcal{E}(D)$ is always a chain graph with chordal[6] chain components [7] [11].

The $d$-separation criterion can be used to identify the skeleton and all the immoralities of the underlying causal DAG [9]. Additional edges can be identified using the fact that the underlying DAG is acyclic and there are no more immoralities. Meek derived 3 local rules (*Meek rules*), introduced in [12], to be recursively applied to identify every such additional edge (see Theorem 3 of [13]). The repeated application of *Meek rules* on this partially directed graph with identified immoralities until they can no longer be used yields the essential graph.

## 2.2 Interventions and Active Learning

Given a set of variables $V = \{x_1, ..., x_n\}$, an *intervention* on a set $S \subset X$ of the variables is an experiment where the performer forces each variable $s \in S$ to take the value of another independent (from other variables) variable $u$, i.e., $s = u$. This operation, and how it affects the joint distribution is formalized by the *do* operator by Pearl [1]. An intervention modifies the causal DAG $D$ as follows: The post intervention DAG $D_{\{S\}}$ is obtained by removing the connections of nodes in $S$ to their parents. The *size of an intervention* $S$ is the number of intervened variables, i.e., $|S|$. Let $S^c$ denote the complement of the set $S$.

CI-based learning algorithms can be applied to $D_{\{S\}}$ to identify the set of removed edges, i.e. parents of $S$ [9], and the remaining adjacent edges in the original skeleton are declared to be the children. Hence,

(R0) The orientations of the edges of the cut between $S$ and $S^c$ in the original DAG $D$ can be inferred.

Then, 4 local Meek rules (introduced in [12]) are repeatedly applied to the original DAG $D$ with the new directions learnt from the cut to learn more till no more directed edges can be identified. Further application of CI-based algorithms on $D$ will reveal no more information. The Meek rules are given below:

(R1) $(a - b)$ is oriented as $(a \rightarrow b)$ if $\exists c$ s.t. $(c \rightarrow a)$ and $(c, b) \notin E$.
(R2) $(a - b)$ is oriented as $(a \rightarrow b)$ if $\exists c$ s.t. $(a \rightarrow c)$ and $(c \rightarrow b)$.
(R3) $(a - b)$ is oriented as $(a \rightarrow b)$ if $\exists c, d$ s.t. $(a - c), (a - d), (c \rightarrow b), (d \rightarrow b)$ and $(c, d) \notin E$.

(R4) $(a - c)$ is oriented as $(a \rightarrow c)$ if $\exists b, d$ s.t. $(b \rightarrow c), (a - d), (a - b), (d \rightarrow b)$ and $(c, d) \notin E$.

The concepts of essential graphs and Markov equivalence classes are extended in [14] to incorporate the role of interventions: Let $\mathcal{I} = \{I_1, I_2, ..., I_m\}$, be a set of interventions and let the above process be followed after each intervention. Interventional Markov equivalence class ($\mathcal{I}$ equivalence) of a DAG is the set of DAGs that represent the same set of probability distributions obtained when the above process is applied after every intervention in $\mathcal{I}$. It is denoted by $[D]_{\mathcal{I}}$. Similar to the observational case, $\mathcal{I}$ *essential graph* of a DAG $D$ is the graph union of all DAGs in the same $\mathcal{I}$ equivalence class; it is denoted by $\mathcal{E}_{\mathcal{I}}(D)$. We have the following sequence:

$$D \rightarrow \text{CI learning} \rightarrow \text{Meek rules} \rightarrow \mathcal{E}(D) \rightarrow I_1 \xrightarrow{a} \text{learn by R0} \xrightarrow{b} \text{Meek rules}$$
$$\rightarrow \mathcal{E}_{\{I_1\}}(D) \rightarrow I_2 \ldots \rightarrow \mathcal{E}_{\{I_1, I_2\}}(D) \ldots \qquad (1)$$

Therefore, after a set of interventions $\mathcal{I}$ has been performed, the essential graph $\mathcal{E}_{\mathcal{I}}(D)$ is a graph with some oriented edges that captures all the causal relations we have discovered so far, using $\mathcal{I}$. Before any interventions happened $\mathcal{E}(D)$ captures the initially known causal directions. It is known that $\mathcal{E}_{\mathcal{I}}(D)$ is a chain graph with chordal chain components. Therefore when all the directed edges are removed, the graph becomes a set of disjoint chordal graphs.

## 2.3 Problem Definition

We are interested in the following question:

**Problem 1.** *Given that all interventions in $\mathcal{I}$ are of size at most $k < n/2$ variables, i.e., for each intervention I, $|I| \leq k, \forall I \in \mathcal{I}$, minimize the number of interventions $|\mathcal{I}|$ such that the partially directed graph with all directions learned so far $\mathcal{E}_{\mathcal{I}}(D) = D$.*

The question is the design of an algorithm that computes the small set of interventions $\mathcal{I}$ given $\mathcal{E}(D)$. Note, of course, that the unknown directions of the edges $D$ are not available to the algorithm. One can view the design of $\mathcal{I}$ as an active learning process to find $D$ from the essential graph $\mathcal{E}(D)$. $\mathcal{E}(D)$ is a chain graph with undirected chordal components and it is known that interventions on one chain components do not affect the discovery process of directed edges in the other components [15]. So we will assume that $\mathcal{E}(D)$ is undirected and a chordal graph to start with. Our notion of algorithm does not consider the time complexity (of statistical algorithms involved) of steps $a$ and $b$ in (1). Given $m$ interventions, we only consider efficiently computing $I_{m+1}$ using (possibly) the graph $\mathcal{E}_{\{I_1, ... I_m\}}$. We consider the following three classes of algorithms:

1. *Non-adaptive algorithm:* The choice of $\mathcal{I}$ is fixed prior to the discovery process.
2. *Adaptive algorithm:* At every step $m$, the choice of $I_{m+1}$ is a deterministic function of $\mathcal{E}_{\{I_1, ... I_m\}}(D)$.
3. *Randomized adaptive algorithm:* At every step $m$, the choice of $I_{m+1}$ is a random function of $\mathcal{E}_{\{I_1, ... I_m\}}(D)$.

The problem is different for complete graphs versus more general chordal graphs since rule R1 becomes applicable when the graph is not complete. Thus we give a separate treatment for each case. First, we provide algorithms for all three cases for learning the directions of complete graphs $\mathcal{E}(D) = K_n$ (undirected complete graph) on $n$ vertices. Then, we generalize to chordal graph skeletons and provide a novel adaptive algorithm with upper and lower bounds on its performance.

The missing proofs of the results that follow can be found in the Appendix.

## 3 Complete Graphs

In this section, we consider the case where the skeleton we start with, i.e. $\mathcal{E}(D)$, is an undirected complete graph (denoted $K_n$). It is known that at any stage in (1) starting from $\mathcal{E}(D)$, rules R1, R3 and R4 do not apply. Further, the underlying DAG $D$ is a directed clique. The directed clique is characterized by an ordering $\sigma$ on $[1 : n]$ such that, in the subgraph induced by $\sigma(i), \sigma(i + 1) \ldots \sigma(n)$, $\sigma(i)$ has no incoming edges. Let $D$ be denoted by $\vec{K}_n(\sigma)$ for some ordering $\sigma$. Let $[1 : n]$ denote the set $\{1, 2 \ldots n\}$. We need the following results on a separating system for our first result regarding adaptive and non-adaptive algorithms for a complete graph.

## 3.1 Separating System

**Definition 1.** *[16, 17] An $(n,k)$-separating system on an $n$ element set $[1:n]$ is a set of subsets $\mathcal{S} = \{S_1, S_2 \ldots S_m\}$ such that $|S_i| \leq k$ and for every pair $i, j$ there is a subset $S \in \mathcal{S}$ such that either $i \in S$, $j \notin S$ or $j \in S$, $i \notin S$. If a pair $i, j$ satisfies the above condition with respect to $\mathcal{S}$, then $\mathcal{S}$ is said to separate the pair $i, j$. Here, we consider the case when $k < n/2$*

In [16], Katona gave an $(n,k)$-separating system together with a lower bound on $|\mathcal{S}|$. In [17], Wegener gave a simpler argument for the lower bound and also provided a tighter upper bound than the one in [16]. In this work, we give a different construction below where the separating system size is at most $\lceil \log_{\lceil n/k \rceil} n \rceil$ larger than the construction of Wegener. However, our construction has a simpler description.

**Lemma 1.** *There is a labeling procedure that produces distinct $\ell$ length labels for all elements in $[1:n]$ using letters from the integer alphabet $\{0, 1 \ldots a\}$ where $\ell = \lceil \log_a n \rceil$. Further, in every digit (or position), any integer letter is used at most $\lceil n/a \rceil$ times.*

Once we have a set of $n$ string labels as in Lemma 1, our separating system construction is straightforward.

**Theorem 1.** *Consider an alphabet $\mathcal{A} = [0 : \lceil \frac{n}{k} \rceil]$ of size $\lceil \frac{n}{k} \rceil + 1$ where $k < n/2$. Label every element of an $n$ element set using a distinct string of letters from $\mathcal{A}$ of length $\ell = \lceil \log_{\lceil \frac{n}{k} \rceil} n \rceil$ using the procedure in Lemma 1 with $a = \lceil \frac{n}{k} \rceil$. For every $1 \leq i \leq \ell$ and $1 \leq j \leq \lceil \frac{n}{k} \rceil$, choose the subset $S_{i,j}$ of vertices whose string's $i$-th letter is $j$. The set of all such subsets $\mathcal{S} = \{S_{i,j}\}$ is a $k$-separating system on $n$ elements and $|\mathcal{S}| \leq (\lceil \frac{n}{k} \rceil) \lceil \log_{\lceil \frac{n}{k} \rceil} n \rceil$.*

## 3.2 Adaptive algorithms: Equivalence to a Separating System

Consider any non-adaptive algorithm that designs a set of interventions $\mathcal{I}$, each of size at most $k$, to discover $\vec{K}_n(\sigma)$. $\mathcal{I}$ has to be a separating system in the worst case over all $\sigma$. This is already known. Now, we prove the necessity of a separating system for deterministic adaptive algorithms in the worst case.

**Theorem 2.** *Let there be an adaptive deterministic algorithm $A$ that designs the set of interventions $\mathcal{I}$ such that the final graph learnt $\mathcal{E}_{\mathcal{I}}(D) = \vec{K}_n(\sigma)$ for any ground truth ordering $\sigma$ starting from the initial skeleton $\mathcal{E}(D) = K_n$. Then, there exists a $\sigma$ such that $A$ designs an $\mathcal{I}$ which is a separating system.*

The theorem above is independent of the individual intervention sizes. Therefore, we have the following theorem, which is a direct corollary of Theorem 2:

**Theorem 3.** *In the worst case over $\sigma$, any adaptive or a non-adaptive deterministic algorithm on the DAG $\vec{K}_n(\sigma)$ has to be such that $\frac{n}{k} \log_{\frac{ne}{k}} n \leq |\mathcal{I}|$. There is a feasible $\mathcal{I}$ with $|\mathcal{I}| \leq \lceil (\frac{n}{k}) - 1) \lceil \log_{\lceil \frac{n}{k} \rceil} n \rceil$*

*Proof.* By Theorem 2, we need a separating system in the worst case and the lower and upper bounds are from [16, 17]. □

## 3.3 Randomized Adaptive Algorithms

In this section, we show that that total number of variable accesses to fully identify the complete causal DAG is $\Omega(n)$.

**Theorem 4.** *To fully identify a complete causal DAG $\vec{K}_n(\sigma)$ on $n$ variables using size-$k$ interventions, $\frac{n}{2k}$ interventions are necessary. Also, the total number of variables accessed is at least $\frac{n}{2}$.*

The lower bound in Theorem 4 is information theoretic. We now give a randomized algorithm that requires $O(\frac{n}{k} \log \log k)$ experiments in expectation. We provide a straightforward generalization of [5], where the authors gave a randomized algorithm for unbounded intervention size.

**Theorem 5.** *Let $\mathcal{E}(D)$ be $K_n$ and the experiment size $k = n^r$ for some $0 < r < 1$. Then there exists a randomized adaptive algorithm which designs an $\mathcal{I}$ such that $\mathcal{E}_{\mathcal{I}}(D) = D$ with probability polynomial in $n$, and $|\mathcal{I}| = \mathcal{O}(\frac{n}{k} \log \log(k))$ in expectation.*

# 4 General Chordal Graphs

In this section, we turn to interventions on a general DAG $G$. After the initial stages in (1), $\mathcal{E}(G)$ is a chain graph with chordal chain components. There are no further immoralities throughout the graph. In this work, we focus on one of the chordal chain components. Thus the DAG $D$ we work on is assumed to be a directed graph with no immoralities and whose skeleton $\mathcal{E}(D)$ is chordal. We are interested in recovering $D$ from $\mathcal{E}(D)$ using interventions of size at most $k$ following (1).

## 4.1 Bounds for Chordal skeletons

We provide a lower bound for both adaptive and non-adaptive deterministic schemes for a chordal skeleton $\mathcal{E}(D)$. Let $\chi(\mathcal{E}(D))$ be the coloring number of the given chordal graph. Since, chordal graphs are perfect, it is the same as the clique number.

**Theorem 6.** *Given a chordal $\mathcal{E}(D)$, in the worst case over all DAGs $D$ (which has skeleton $\mathcal{E}(D)$ and no immoralities), if every intervention is of size at most $k$, then $|\mathcal{I}| \geq \frac{\chi(\mathcal{E}(D))}{k} \log_{\frac{\chi(\mathcal{E}(D))e}{k}} \chi(\mathcal{E}(D))$ for any adaptive and non-adaptive algorithm with $\mathcal{E}_{\mathcal{I}}(D) = D$.*

*Upper bound:* Clearly, the separating system based algorithm of Section 3 can be applied to the vertices in the chordal skeleton $\mathcal{E}(D)$ and it is possible to find all the directions. Thus, $|\mathcal{I}| \leq \frac{n}{k} \log_{\lceil \frac{n}{k} \rceil} n \leq \frac{\alpha(\mathcal{E}(D))\chi(\mathcal{E}(D))}{k} \log_{\lceil \frac{n}{k} \rceil} n$. This with the lower bound implies an $\alpha$ approximation algorithm (since $\log_{\lceil \frac{n}{k} \rceil} n \leq \log_{\frac{\chi(\mathcal{E}(D))e}{k}} \chi(\mathcal{E}(D))$, under a mild assumption $\chi(\mathcal{E}(D)) \leq \frac{n}{e}$).

**Remark:** The separating system on $n$ nodes gives an $\alpha$ approximation. However, the new algorithm in Section 4.3 exploits chordality and performs much better empirically. It is possible to show that our heuristic also has an $\alpha$ approximation guarantee but we skip that.

## 4.2 Two extreme counter examples

We provide two classes of chordal skeletons $G$: One for which the number of interventions close to the lower bound is *sufficient* and the other for which the number of interventions *needed* is very close to the upper bound.

**Theorem 7.** *There exists chordal skeletons such that for any algorithm with intervention size constraint $k$, the number of interventions $|\mathcal{I}|$ required is at least $\alpha \frac{(\chi-1)}{2k}$ where $\alpha$ and $\chi$ are the independence number and chromatic numbers respectively. There exists chordal graph classes such that $|\mathcal{I}| = \lceil \frac{\chi}{k} \rceil \lceil \log_{\lceil \frac{\chi}{k} \rceil} \chi \rceil$ is sufficient.*

## 4.3 An Improved Algorithm using Meek Rules

In this section, we design an adaptive deterministic algorithm that *anticipates* Meek rule R1 usage along with the idea of a separating system. We evaluate this experimentally on random chordal graphs. First, we make a few observations on learning connected directed trees $T$ from the skeleton $\mathcal{E}(T)$ (undirected trees are chordal) that do not have immoralities using Meek rule R1 where every intervention is of size $k = 1$. Because the tree has no cycle, Meek rules R2-R4 do not apply.

**Lemma 2.** *Every node in a directed tree with no immoralities has at most one incoming edge. There is a root node with no incoming edges and intervening on that node alone identifies the whole tree using repeated application of rule R1.*

**Lemma 3.** *If every intervention in $\mathcal{I}$ is of size at most 1, learning all directions on a directed tree $T$ with no immoralities can be done adaptively with at most $|\mathcal{I}| \leq O(\log_2 n)$ where $n$ is the number of vertices in the tree. The algorithm runs in time $\text{poly}(n)$.*

**Lemma 4.** *Given any chordal graph and a valid coloring, the graph induced by any two color classes is a forest.*

In the next section, we combine the above single intervention adaptive algorithm on directed trees which uses Meek rules, with that of the non-adaptive separating system approach.

#### 4.3.1 Description of the algorithm

The key motivation behind the algorithm is that, a pair of color classes is a forest (Lemma 4). Choosing the right node to intervene leaves only a small subtree unlearnt as in the proof of Lemma 3. In subsequent steps, suitable nodes in the remaining subtrees could be chosen until all edges are learnt. We give a brief description of the algorithm below.

Let $G$ denote the initial undirected chordal skeleton $\mathcal{E}(D)$ and let $\chi$ be its coloring number. Consider a $(\chi, k)$ separating system $\mathcal{S} = \{S_i\}$. To intervene on the actual graph, an intervention set $I_i$ corresponding to $S_i$ is chosen. We would like to intervene on a node of color $c \in S_i$.

Consider a node $v$ of color $c$. Now, we attach a score $P(v, c)$ as follows. For any color $c' \notin S_i$, consider the induced forest $F(c, c')$ on the color classes $c$ and $c'$ in $G$. Consider the tree $T(v, c, c')$ containing node $v$ in $F$. Let $d(v)$ be the degree of $v$ in $T$. Let $T_1, T_2, \ldots T_{d(v)}$ be the resulting disjoint trees after node $v$ is removed from $T$. If $v$ is intervened on, according to the proof of Lemma 3: a) All edge directions in all trees $T_i$ except one of them would be learnt when applying Meek Rules and rule R0. b) All the directions from $v$ to all its neighbors would be found.

The score is taken to be the total number of edge directions guaranteed to be learnt in the worst case. Therefore, the score $P(v)$ is: $P(v) = \sum_{c':|c,c' \bigcap|=1} \left( |T(c, c')| - \max_{1 \le j \le d(v)} |T_j| \right)$. The node with the highest score among the color class $c$ is used for the intervention $I_i$. After intervening on $I_i$, all the edges whose directions are known through Meek Rules (by repeated application till nothing more can be learnt) and R0 are deleted from $G$. Once $\mathcal{S}$ is processed, we recolor the sparser graph $G$. We find a new $\mathcal{S}$ with the new chromatic number on $G$ and the above procedure is repeated. The exact hybrid algorithm is described in Algorithm 1.

**Theorem 8.** *Given an undirected choral skeleton $G$ of an underlying directed graph with no immoralities, Algorithm 1 ends in finite time and it returns the correct underlying directed graph. The algorithm has runtime complexity polynomial in $n$.*

---

**Algorithm 1** Hybrid Algorithm using Meek rules with separating system

---

1: **Input:** Chordal Graph skeleton $G = (V, E)$ with no Immoralities.
2: Initialize $\vec{G}(V, E_d = \emptyset)$ with $n$ nodes and no directed edges. Initialize time $t = 1$.
3: **while** $E \ne \emptyset$ **do**
4:      Color the chordal graph $G$ with $\chi$ colors.   ▷ Standard algorithms exist to do it in linear time
5:      Initialize color set $\mathcal{C} = \{1, 2 \ldots \chi\}$. Form a $(\chi, \min(k, \lceil \chi/2 \rceil))$ separating system $\mathcal{S}$ such that $|S| \le k, \forall S \in \mathcal{S}$.
6:      **for** $i = 1$ until $|\mathcal{S}|$ **do**
7:          Initialize Intervention $I_t = \emptyset$.
8:          **for** $c \in S_i$ and every node $v$ in color class $c$ **do**
9:              Consider $F(c, c')$, $T(c, c', v)$ and $\{T_j\}_1^{d(i)}$ (as per definitions in Sec. 4.3.1).
10:              Compute: $P(v, c) = \sum_{c' \in \mathcal{C} \bigcap S_i^c} |T(c, c', v)| - \max_{1 \le j \le d(i)} |T_j|$.
11:          **end for**
12:          **if** $k \le \chi/2$ **then**
13:              $I_t = I_t \bigcup \{ \underset{c \in S_i}{} \underset{v: P(v,c) \ne 0}{\arg\max} P(v, c)\}$.
14:          **else**
15:              $I_t = I_t \cup_{c \in S_i} \{$First $\frac{k}{\lceil \chi/2 \rceil}$ nodes $v$ with largest nonzero $P(v, c)\}$.
16:          **end if**
17:          $t = t + 1$
18:          Apply R0 and Meek rules using $E_d$ and $E$ after intervention $I_t$. Add newly learnt directed edges to $E_d$ and delete them from $E$.
19:      **end for**
20:      Remove all nodes which have degree 0 in G.
21: **end while**
22: **return** $\vec{G}$.

---

# 5 Simulations

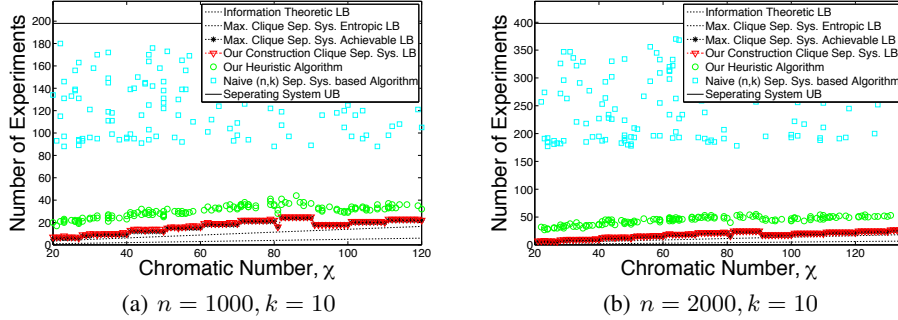

(a) $n = 1000, k = 10$        (b) $n = 2000, k = 10$

Figure 1: $n$: no. of vertices, $k$: Intervention size bound. The number of experiments is compared between our heuristic and the naive algorithm based on the $(n, k)$ separating system on random chordal graphs. The red markers represent the sizes of $(\chi, k)$ separating system. Green circle markers and the cyan square markers for the same $\chi$ value correspond to the number of experiments required by our heuristic and the algorithm based on an $(n, k)$ separating system(Theorem 1), respectively, on the same set of chordal graphs. Note that, when $n = 1000$ and $n = 2000$, the naive algorithm requires on average about 130 and 260 (close to $n/k$) experiments respectively, while our algorithm requires at most $\sim 40$ (orderwise close to $\chi/k = 10$) when $\chi = 100$.

We simulate our new heuristic, namely Algorithm 1, on randomly generated chordal graphs and compare it with a naive algorithm that follows the intervention sets given by our $(n, k)$ separating system as in Theorem 1. Both algorithms apply R0 and Meek rules after each intervention according to (1). We plot the following lower bounds: a) *Information Theoretic LB* of $\frac{\chi}{2k}$ b) *Max. Clique Sep. Sys. Entropic LB* which is the chromatic number based lower bound of Theorem 6. Moreover, we use two known $(\chi, k)$ separating system constructions for the maximum clique size as "references": The best known $(\chi, k)$ separating system is shown by the label *Max. Clique Sep. Sys. Achievable LB* and our new simpler separating system construction (Theorem 1) is shown by *Our Construction Clique Sep. Sys. LB*. As an upper bound, we use the size of the best known $(n, k)$ separating system (without any Meek rules) and is denoted *Separating System UB*.

*Random generation of chordal graphs:* Start with a random ordering $\sigma$ on the vertices. Consider every vertex starting from $\sigma(n)$. For each vertex $i$, $(j, i) \in E$ with probability inversely proportional to $\sigma(i)$ for every $j \in S_i$ where $S_i = \{v : \sigma^{-1}(v) < \sigma^{-1}(i)\}$. The proportionality constant is changed to adjust sparsity of the graph. After all such $j$ are considered, make $S_i \cap \text{ne}(i)$ a clique by adding edges respecting the ordering $\sigma$, where $\text{ne}(i)$ is the neighborhood of $i$. The resultant graph is a DAG and the corresponding skeleton is chordal. Also, $\sigma$ is a perfect elimination ordering.

**Results:** We are interested in comparing our algorithm and the naive one which depends on the $(n, k)$ separating system to the size of the $(\chi, k)$ separating system. The size of the $(\chi, k)$ separating system is roughly $\tilde{O}(\chi/k)$. Consider values around $\chi = 100$ on the x-axis for the plots with $n = 1000, k = 10$ and $n = 2000, k = 10$. Note that, our algorithm performs very close to the size of the $(\chi, k)$ separating system, i.e. $\tilde{O}(\chi/k)$. In fact, it is always $< 40$ in both cases while the average performance of naive algorithm goes from 130 (close to $n/k = 100$) to 260 (close to $n/k = 200$). The result points to this: For random chordal graphs, the structured tree search allows us to learn the edges in a number of experiments quite close to the lower bound based only on the maximum clique size and not $n$. The plots for $(n, k) = (500, 10)$ and $(n, k) = (2000, 20)$ are given in Appendix.

## Acknowledgments

Authors acknowledge the support from grants: NSF CCF 1344179, 1344364, 1407278, 1422549 and a ARO YIP award (W911NF-14-1-0258). We also thank Frederick Eberhardt for helpful discussions.

## Footnotes

[1]A separating system is a 0-1 matrix with $n$ distinct columns and each row has at most $k$ ones.

[2]Given Bayesian network, any CI relation implied by d-separation holds true. All the CIs implied by the distribution can be found using d-separation if the distribution is faithful. Faithfulness is a widely accepted assumption, since it is known that only a measure zero set of distributions are not faithful [10].

[3]Skeleton of a DAG is the undirected graph obtained when directed edges are converted to undirected edges.

[4]An induced subgraph on $X, Y, Z$ is an immorality if $X$ and $Y$ are disconnected, $X \rightarrow Z$ and $Z \leftarrow Y$.

[5]Graph union of two DAGs $D_1 = (V, E_1)$ and $D_2 = (V, E_2)$ with the same skeleton is a partially directed graph $D = (V, E)$, where $(v_a, v_b) \in E$ is undirected if the edges $(v_a, v_b)$ in $E_1$ and $E_2$ have different directions, and directed as $v_a \rightarrow v_b$ if the edges $(v_a, v_b)$ in $E_1$ and $E_2$ are both directed as $v_a \rightarrow v_b$.

[6]An undirected graph is chordal if it has no induced cycle of length greater than 3.

[7]This means that $\mathcal{E}(D)$ can be decomposed as a sequence of undirected chordal graphs $G_1, G_2 \ldots G_m$ (chain components) such that there is a directed edge from a vertex in $G_i$ to a vertex in $G_j$ only if $i < j$

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
