[Supplementary Material · Supplement-active-new.pdf]

# Appendix

## 5.1 Proof of Lemma 1

We describe a string labeling procedure as follows to label elements of the set $[1:n]$.

*String Labelling:* Let $a > 1$ be a positive integer. Let $x$ be the integer such that $a^x < n \le a^{x+1}$. $x + 1 = \lceil \log_a n \rceil$. Every element $j \in [1:n]$ is given a label $L(j)$ which is a string of integers of length $x + 1$ drawn from the alphabet $\{0, 1, 2 \ldots a\}$ of size $a + 1$. Let $n = p_d a^d + r_d$ and $n = p_{d-1}a^{d-1} + r_{d-1}$ for some integers $p_d, p_{d-1}, r_d, r_{d-1}$, where $r_d < a^d$ and $r_{d-1} < a^{d-1}$. Now, we describe the sequence of the $d$-th digit across the string labels of all elements from 1 to $n$:

1. Repeat 0 $a^{d-1}$ times, repeat the next integer 1 $a^{d-1}$ times and so on circularly [8] from $\{0, 1 \ldots a - 1\}$ till $p_d a^d$.

2. After that, repeat 0 $\lceil r_d/a \rceil$ times followed by 1 $\lceil r_d/a \rceil$ times till we reach the $n$th position. Clearly, $n$-th integer in the sequence would not exceed $a - 1$.

3. Every integer occurring *after* the position $a^{d-1}p_{d-1}$ is increased by 1.

From the three steps used to generate every digit, a straightforward calculation shows that every integer letter is repeated at most $\lceil n/a \rceil$ times in every digit $i$ in the string. Now, we would like to prove inductively that the labels are distinct for all $n$ elements. Let us assume the induction hypothesis: For all $n < a^{q+1}$, the labels are distinct. The base case of $q = 0$ is easy to see. Then, we would like to show that for $a^{q+1} \le n < a^{q+2}$, the labels are distinct.

Another way of looking at the labeling procedure is as follows. Let $n = a^{q+1}p + r$ with $r < a^{q+1}$. Divide the label matrix $L$ (of dimensions $(q + 2) \times n$) into two parts, one $L_1$ consisting of the first $pa^{q+1}$ columns and the other $L_2$ consisting of the remaining columns. The first $q + 1$ rows of $L_1$ is nothing but the string labels for all numbers from 0 to $pa^{q+1}$ expressed in base $a$. For any row $i \le \lceil \log_a r \rceil$ in the original matrix $L$ of labels, till the end of first $pa^{q+1}$ columns, the labeling procedure would be still in Step 1. After that, one can take $r$ to be the new size of the set of elements to be labelled and then restart the procedure with this $r$. Therefore we have the following *key* observation: $L_2(1 : \lceil \log_a r \rceil, :)$ (the matrix with first $\lceil \log_a r \rceil$ rows of $L_2$) is nothing but the label matrix for $r$ distinct elements from the above labeling procedure.

Since, $r < a^{q+1}$, by the induction hypothesis, the columns are distinct. Hence, any two columns in $L_2$ are distinct. Suppose the first $q + 1$ rows of two columns $b$ and $c$ of $L_1$ are identical. These correspond to base $a$ expansion of $b - 1$ and $c - 1$. They are separated by at least $a^{q+1} + 1$ columns. But the last row of columns $b$ and $p$ in $L_1$ has to be distinct because according to Step 2 and Step 3 of the labeling procedure, in the $q + 2^{th}$ row, every integer is repeated at most $\lceil n/a \rceil \le a^{q+1}$ times continuously, and only once. Therefore, any two columns in $L_1$ are distinct. The last row entries in $L_1$ are different from $L_2$ because of the addition in Step 3. Therefore, all columns of $L$ are distinct. Hence, by induction, the result is shown.

## 5.2 Proof of Theorem 1

By Lemma 1, $i$th place has at most $\lceil \frac{n}{\lceil n/k \rceil} \rceil \le k$ occurrences of symbol $j$. Therefore, $|S_{i,j}| \le k$. Now, consider the pair of distinct elements $p, q \in [1:n]$. Since they are labelled distinctly (Lemma 1), there is at least one letter $i$ in their string labels where they differ. Suppose the distinct $i$th letters are $a, b \in \mathcal{A}$, $a \ne b$ and let us say $a \ne 0$ without loss of generality. Then, clearly the separation criterion is met by the subset $S_{i,a}$. This proves the claim.

## 5.3 Proof of Theorem 2

We construct a worst case $\sigma$ inductively. Before every step $m$, the adaptive algorithm deterministically chooses $I_m$ based on $\mathcal{E}_{\{I_1, I_2 \ldots I_{m-1}\}}(K_n)$. Therefore, we will reveal a partial order $\sigma^{(m-1)}$ to satisfy the observations so far. Inductively for every $m$, we will make sure that after $I_m$ is chosen by the algorithm, further details about $\sigma$ can be revealed to form $\sigma^{(m)}$ such that after intervening on $I_2$

and then applying R0, we will make sure there is no opportunity to apply the rule $R2$. This would make sure that $\mathcal{I}$ is a separating system on $n$ elements.

Before intervention at any step $m$, let us 'tag' every vertex $i$ using a subset $C_i^{(m-1)} \subseteq [1:m]$ such that $C_i^{(m-1)} = \{p : i \in I_p, \ p \leq m-1\}$. $C_i^{(m-1)}$ contains indices of all those interventions that contain vertex $i$ before step $m$. Let $\mathcal{C}^{(m-1)}$ contain distinct elements of the multi-set $\{C_i^{(m-1)}\}$. We will construct $\sigma$ partially such that it satisfies the following criterion always:

*Inductive Hypothesis:* The partial order $\sigma^{(m-1)}$ is such that for any two elements $i, j$ with $C_i$ and $C_j$, $i$ and $j$ are incomparable if $C_i = C_j$ and comparable otherwise. This means the edges between the elements tagged with the same tag $C$ has not been revealed, and thus the relevant directed edges are not known by the algorithm.

Now, we briefly digress to argue that if we could construct $\sigma^{(1)}, \sigma^{(2)} \ldots$ satisfying such a property throughout, then clearly all vertices must be tagged differently otherwise the directions among the vertices that are tagged similarly cannot be learned by the algorithm. Therefore, the algorithm has not succeeded in its task. If all vertices are tagged differently, then it means it is a separating system.

*Construction of $\sigma^{(m)}$:* We now construct $\sigma^{(m)}$ that can be shown to satisfy the induction hypothesis before step $m + 1$. Before step $m$, consider the vertices in $C \in \mathcal{C}^{(m-1)}$ for any $C$. Let the current intervention be $I_m$ chosen by the deterministic algorithm. We make the following changes: Modify $\sigma^{(m-1)}$ such that vertices in $I_m \bigcap C$ come before $(I_m)^c \bigcap C$ in the partial order $\sigma^{(m)}$ (vertices inside either sets are still not ordered amongst themselves) in the ordering and clearly the directions between these two sets are revealed by R0. By the induction hypothesis for step $m$ and with the new tagging of vertices into $\mathcal{C}^{(m)}$, it is easy to see that only directions between distinct $C's$ in the new $\mathcal{C}^{(m)}$ have been revealed and all directions within a tag set $C$ are not revealed and all vertices in a tag set are contiguous in the ordering so far. We need to only show that rule R2 cannot reveal anymore edges amongst vertices in $C \in \mathcal{C}^{(m)}$ after the new $\sigma^{(m)}$ and intervention $I_m$. Suppose there are two vertices $i, j$ such that just after intervention $I_m$ and the modified $\sigma^{(m)}$, they are tagged identically and application of R2 reveals the direction between $i$ and $j$ before the next intervention. Then there has to be a vertex $k$ tagged differently from $i, j$ such that $j \to k$ and $k \to i$ are both known. But this implies that $j$ and $i$ are comparable in $\sigma^{(m)}$ leading to a contradiction. This implies the hypothesis holds for step $m + 1$.

*Base case:* Trivially, the induction hypothesis holds for step $0$ where $\sigma^{(0)}$ leaves the entire set unordered.

## 5.4 Proof of Lemma 2

The proof is a direct obvious consequence of acyclicity, non-existence of immoralities and the definition of rule R1.

## 5.5 Proof of Lemma 3

By Lemma 2, it is sufficient for an algorithm to identify the root node of the tree. Suppose the root node is $b$ unknown to the algorithm. Every tree has a single vertex separator that partitions the tree into components each of which has size at most $\frac{2}{3}n$ [18]. Choose that vertex separator $a_1$ (it can be found in by removing every node and determining the components left). If it is a root node we stop here. Otherwise, its parent $p_1$ (if it is not) after application of rule R0 is identified. Let us consider component trees $T_1, T_2 \ldots T_k$ that result by removing node $a_1$. Let $T_1$ contain $p_1$. All directions in all other trees are known after repeated application of R1 on the original tree after R0 is applied. Directions in T1 will not be known. For the next step, $\mathcal{E}(T_1)$ is the new skeleton which has no immoralities. Again, we find the best vertex separator $a_2$ and the process continues. This procedure will terminate at some step $j$ when $a_j = b$ or there is only one node left which should be $b$ by Lemma 2. Since the number of nodes reduce by about $1/3$ at least each time, and initially it can be at most $n$, this procedure terminates in at most $O(\log_2 n)$ steps.

## 5.6 Proof of Lemma 4

The graph induced by two colors classes in any graph is a bi-partite graph and bi-partite graphs do not have odd induced cycles. Since the graph and any induced subgraph is chordal, it implies the induced graph on a pair of color classes does not have a cycle. This proves the theorem.

## 5.7 Proof of Theorem 4

Assume $n$ is even for simplicity. We define a family of partial order $\sigma^{(p)}$ as follows: Group $i, i+1$ into $C_i$. Ordering among $i$ and $i+1$ is not revealed. But all the edges between $C_i$ and $C_j$ for any $j > i$ are directed from $C_i$ to $C_j$. Now, one has to design a set of interventions such that exactly one node among every $C_i$ is intervened on at least once. This is because, if neither $i$ nor $i+1$ in $C_i$ are intervened on, then the direction between $i$ and $i+1$ cannot be figured out by applying rule R2 on any other set of directions in the rest of the graph. Since the size of every intervention is at most $k$ and at least $n/2$ nodes need to be covered by intervention sets, the number of interventions required is at least $\frac{n}{2k}$.

## 5.8 Proof of Theorem 5

*Proof.* Separate $n$ vertices arbitrarily into $\frac{n}{k}$ disjoint subsets $C_i$ of size-$k$. Let the first $n/k$ interventions $\{I_1, I_2, ..., I_{n/k}\}$ be such that $I_i(v) = 1$ if and only if $v \in C_i$. This divides the problem of learning a clique of size $n$ into learning $n/k$ cliques of size $k$. Then, we can apply the clique learning algorithm in [5] as a black box to each of the $\frac{n}{k}$ blocks: Each block is learned with probability $k^{-c}$ after $\log c \log k$ experiments in expectation. For $k = cn^r$, choose $c > 1/r - 1$. Then the union bound over $n/k$ blocks yields probability polynomial in $n$. Since each block takes $\mathcal{O}(\log \log k)$ experiments, we need $\frac{n}{k}\mathcal{O}(\log \log k)$ experiments. $\qquad\square$

## 5.9 Proof of Theorem 6

We need the following definitions and some results before proving the theorem.

**Definition 2.** *A perfect elimination ordering $\sigma_p = \{v_1, v_2 \ldots v_n\}$ on the vertices of an undirected chordal graph $G$ is such that for all $i$, the induced neighborhood of $v_i$ on the subgraph formed by $\{v_1, v_2 \ldots v_{i-1}\}$ is a clique.*

**Lemma 5.** *( [2]) If all directions in the chordal graph are according to perfect elimination ordering (edges go only from vertices lower in the order to higher in the order), then there are no immoralities.*

We make the following observation: Let the directions in a graph $D$ be oriented according to an ordering $\sigma$ on the vertices. If a clique comes first in the ordering, then the knowledge of edge directions in the rest of the graph, excluding that of the clique, cannot help at any stage of the intervention process on the clique; because all the edges are directed outwards from the clique and hence none of the Meek rules apply. This is because, if $a \to b$ is to be inferred by Meek rules from other known directions, then either there has to be a known edge direction into $a$ or $b$ before the inference step. So if one of the directed edges not from the clique was to help in the discovery process, either that edge has to be directed towards $a$ or $b$ (like in Meek rules R1, R2 and R3), or it has to be directed towards $c$ in another $c \to a$ (R4) which belongs to the clique. Both the above cases are not possible.

**Lemma 6.** *( [2]) Let $C$ be a maximum clique of an undirected chordal graph $\mathcal{E}(D)$, then there is an underlying DAG $D$ on the chordal skeleton that is oriented according to a perfect elimination ordering (implying no immoralities), where the clique $C$ occurs first.*

By Lemmas 5, 6 and the observation above, given a chordal skeleton, we can construct a DAG on the skeleton with no immoralities such that the directions of the maximum clique in $D$ cannot be learned by using knowledge of the directions outside. This means that only the intervention sets $\{I_1 \bigcap C, I_2 \bigcap C \ldots\}$ matter for learning the directions on this clique. Therefore inference on the clique is isolated. Hence, all the lower bounds for the clique case transfer to this case and since the size of the largest clique is exactly the coloring number of the chordal skeleton, the theorem follows.

## 5.10 Proof of Theorem 7

*Example with a feasible solution with $|\mathcal{I}|$ close to the lower bound:* Consider a graph $G$ that can be partitioned into a clique of size $\chi$ and an independent set $\alpha$. Such graphs are called split graphs and as $n \to \infty$, the fraction of split graphs to chordal graphs tends to 1. If $\mathcal{E}(D) = G$ where $G$ is a split graph skeleton, it is enough to intervene only on the nodes in the clique and therefore the number of interventions that are needed is that for the clique. It is certainly possible to orient the edges in such a way so as to avoid immoralities, since the graph is chordal.

*Example with $|\mathcal{I}|$ which needs to be close to the upper bound:* We construct a connected chordal skeleton with independent set $\alpha$ and clique size $\chi$ (also coloring number) such that it would require $\frac{\alpha(\chi-1)}{2k}$ interventions at least for any algorithm over a class of orientations.

Consider a line $L$ consisting of vertices $1, 2 \ldots 2\alpha$ such that every node $1 < i < 2\alpha$ is connected to $i-1$ and $i+1$. For, all $1 \le p \le \alpha$, consider a clique $C_p$ of size $\chi$ which only has nodes $2p-1, 2p$ from the line $L$. Now assume that the actual orientation of the L is $1 \to 2 \ldots \to 2\alpha$. In every clique, the orientation is partially specified as follows: In every clique $C_p$, all edges from node $2p-1$ are outgoing. It is very clear that this partial orientation excludes all immoralities. Further, each clique $C_p - \{2p-1\}$ can have any arbitrary orientation out of $\chi-1$ possible ones in the actual DAG. Now, even if all the specified directions are revealed to the algorithm, the algorithm has to intervene on all $\alpha$ disjoint cliques $\{C_p - \{2p-1\}\}_{p=1}^{\alpha}$ each of size $\chi-1$ and directions in one clique will not force directions on the others through any of the Meek rules or rule R0. Therefore, the lower bound of $\frac{\alpha(\chi-1)}{2}$ total node accesses (total number of nodes intervened) is implied by Theorem 4. Given every intervention is of size $k$, these chordal skeletons with the revealed partial order needs at least $\frac{\alpha(\chi-1)}{2k}$ more experiments.

## 5.11 Performance Comparison of Our Algorithm vs. Naive Scheme for $n = 500, k = 10$ and $n = 2000, k = 20$

(a) $n = 500, k = 10$      (b) $n = 2000, k = 20$

Figure 2: $n$: no. of vertices, $k$: Intervention size bound. The number of experiments is compared between our heuristic and the naive algorithm based on the $(n, k)$ separating system on random chordal graphs. The red markers represent the sizes of $(\chi, k)$ separating system. Green circle markers and the cyan square markers for the same $\chi$ value correspond to the number of experiments required by our heuristic and the algorithm based on an $(n, k)$ separating system(Theorem 1), respectively, on the same set of chordal graphs. All four plots (including the ones in the main text) indicate that our algorithm requires number of experiments proportional to the clique number $\chi$, whereas naive separating system based algorithm requires experiments on the order of number of variables $n$.

## 5.12 Proof of Theorem 8

We provide the following justifications for the correctness of Algorithm 1.

1. At line 4 of the algorithm, when Meek rules and R0 are applied after every intervention, the intermediate graph $G$, with unlearned edges, will always be a disjoint union of chordal components (refer to (1) and the comments below) and hence a chordal graph.

2. The number of unlearned edges before and after the main while loop in Algorithm 1 reduces by at least one. Every edge in $E$ is incident on two colors and one of the colors is always picked for processing because we use a separating system on the colors. Therefore, one node belonging to some edge has a positive score and is intervened on. The edge direction is learnt through rule R0. Therefore, the algorithm terminates.

3. It identifies the correct $\vec{G}$ because every edge is inferred after some intervention $I_t$ by applying rule R0 and Meek rules as in (1) both of which are correct.

4. the algorithm has polynomial run time complexity because the main while loop ends in time $|E|$.

## Footnotes

[8]Circular means that after $a - 1$ is completed, we start with 0 again.