[Reviews · NeurIPS 2015]

Submitted by Assigned_Reviewer_1

From the abstract I conclude the authors assume that, for a certain problem with a certain data set, it is possible to learn a causal network, and now the task is to improve upon this by allowing interventions (of limiting size). So the problem of causal inference of the network is assumed (not questioned). This is already a bold assumption which is in many cases not true.

Nevertheless, let's assume this because if the method could accomplish this in this limited framework, it would still be a great result.

Looking into the results section, I cannot understand what the authors are trying to do. In any case it is inadequate to convince me that the proposed method does what is stated.

For a problem of that calibre, the numerical results need to be thoughtfully presented to provide a clear case by studying a non-trivial problem. Without this there is the danger that the proven results hold only in unrealistic scenarios.

Summary: The paper deals with the problem of learning causal networks with interventions. The interventions are constraint in size. The goal is learn a causal network by using a smaller sample size.

Submitted by Assigned_Reviewer_2

This paper considers the problem of causal learning with interventions, where the size of interventions is bounded. By using the concept of separating system, the paper shows a number of new theoretical results on the causal learning with interventions, for complete graphs and chordal graphs with no immoralities. Authors also proposes a new adaptive algorithms for chordal graphs, performing very close to the theoretical LB.

The paper is quite theoretical with many theorems and proof.

Concerns: 1) The structure assumptions on complete graphs and chordal graphs with no immoralities seem very limiting. In practice, the graph structures are usually not complete but more complex than the chordal graphs. In which application or areas can these two models be applied or of potential interest?

2) what would be a good strategy to choose the best k? It seems a natural question following the results of small interventions.

3) Section 4: the claim of intervention on a general DAG G is imprecise due to later assumptions. This statement should be modified.

======== Authors clarified our (minor) questions and we vote for the acceptance of this paper.
Summary: The paper provides new advances on causal learning with interventions, where the size of intervention is bounded and the number of experiments is minimized. A few of new theoretical results are shown for both complete and chordal graphs. Experiments with simulations show their algorithm for the chordal graph approaches to the lower bound.

Submitted by Assigned_Reviewer_3

This paper addresses the problem of learning causal networks with interventions, when each intervention is limited to size k. The paper is generally well-written and addresses a relevant question, as it is generally not feasible to learn the causal structure from observational data alone. Moreover, in some cases, it may also not be possible to perform arbitrarily large interventions, but it is possible to perform interventions over smaller subsets of the variables.

The authors prove a number of results around the number of interventions required to learn complete and chordal graphs, and, while I was not able to check all the proofs in detail, the results are as expected (and appear to be correct).

The results on chordal graphs are applicable to general causal structures in the sense that application of conditional independence learning and Meek rules results in a chain graph with chordal chain components.

In addition, the authors implement a novel adaptive deterministic algorithm and test it on randomly generated chordal graphs.

The simulations show that the algorithm performs close to the lower bound for worst case chordal graphs using adaptive or non-adaptive algorithms.

Additionally, it significantly outperforms the naive separating systems approach for complete graphs. Overall, this is a strong paper that should be accepted. I list below the specific contributions.

- The paper gives a novel separating system construction, where the size is larger than that of Wegener (1979) but has a simpler description. This construction is later used for the results on adaptive and non-adaptive algorithms for a complete graph.

- It had previously been shown that any non-adaptive algorithm needs a (n, k) separating system to learn a complete graph. The authors show that the same is true for any adaptive deterministic algorithm.

- The authors show that to learn a complete causal DAG on n variables using k-size interventions, n/2k interventions are necessary. This bound holds for any algorithm.

- The authors give a straightforward extension of a result by Hu et al. (2014), which showed that randomized adaptive algorithms need only log log n interventions with high probability for the unbounded case for complete graphs. The authors show that O(n/k log log k) interventions suffice with high probability when the interventions are bounded by k in size.

- The authors extend the lower bound for adaptive algorithms for general chordal graphs and show that the number of experiments from a ((G), k) separating system is needed where (G) is the chromatic number of the skeleton graph.

- The authors give two extreme examples; one for which the number of interventions is close to the lower bound is sufficient and the other for which the number of interventions needed is close to the upper bound.

- The authors give a novel adaptive deterministic algorithm that uses the idea of separating systems together with adaptability to Meek rules. They test their algorithm on randomly generated chordal graphs and observe that it performs close to the (, k) separating system and significantly outperforms the (n, k) separating system (for complete graphs).

** Additional Comments: - The "Meek rules" described in Section 2.2 were introduced by Verma and Pearl (1992) and Meek (1995) proved certain properties of them. You should cite Verma92.

- Top of pg. 5, replace "also provided a tighter upper bound in [15]" with "also provided a tighter upper bound than the one in [15] -On pg. 8, the "Information Theoretic LB" is described as /2k.

Where is this bound given?

References: Verma, Thomas, and Pearl, Judea. "An algorithm for deciding if a set of observed independencies has a causal explanation." Proceedings of the Eighth international conference on uncertainty in artificial intelligence. Morgan Kaufmann Publishers Inc., 1992.
Summary: This is a strong paper that presents a number of interesting theoretical results on learning causal graphs with limited size interventions.

Additionally, a novel algorithm for learning is implemented and its effectiveness is demonstrated on simulated randomized chordal graphs.

I believe that this is a strong paper that should be accepted.

Submitted by Assigned_Reviewer_4

This paper concerns on learning causal graphs. Specifically, the paper concerns on a scenario where the sizes of the interventions are bounded and the goal is to find the smallest possible set of such interventions in order to be able to direct all edges.

This paper provides several interesting theoretical results on the topic. Result-wise, the paper is worth of publishing. However, the problem is in the presentation. The paper has too many results to fit in the page limit and thus the presentation very superficial and lots of stuff in gone through quickly. I would recommend that either the paper would concentrate on the most important results and the rest would be cut or that the paper would be published in a journal.

064: by Eberhardt et al. [3], where they provided

100: causal DAG

137: What is S^c?

A separating system is defined both in footnote 1 and definition 1 and the definitions are not equivalent.

Missing definitions: induced cycle, chain graph

260: O(n/k log log k)?

349: chordal skeleton
Summary: The paper has interesting theoretical results. However, there are too many results to be presented in sufficient detail.

Author Feedback
Author rebuttal: [Reviewer 1]
We are aware of these shortcomings due to space requirements. The experiments should ideally be "real experiments" in the sense that the variables should be intervened on. Conditional Independence tests between interventions have computational complexity issues which we do not consider in this work (as also in previous work on this topic). Further, the problem of designing interventions is done assuming conditional independence tests are perfect. We will add a short discussion and conclusion section as suggested.

[Reviewer 2]
The information theoretic lower bound follows from the fact that any undirected chordal graph can in the worst case be directed to start the corresponding PEO from the maximum clique. Then, just to learn this clique, information theoretically (which is proven for the sole clique case in Theorem 4) we need chi/2k experiments. We have not devised a separate section for this straightforward extension of Theorem 4 using the ideas from (the proof of) Theorem 6. We thank the reviewer for the historically accurate references and insightful remarks.

[Reviewer 3]

S^c is complement of the set S. We will add a note to explain this notation.
Both separating system definitions are equivalent. Footnote 1 helps the reader visualize the problem as we did. All other changes have been made- thank you for your input.

[Reviewer 4]

Our paper studies the problem of Causal DAG learning under the standard model of Pearl (SEM model- Structural Equation Models with Independent Noise see ref.1). In this model, it is possible to learn causal DAGs using interventions (i.e. forcing a subset of variables in some specific values). Interventions are frequently used in experimental design in statistics and have numerous applications e.g. medicine and genomics. Prior work (see e.g. [2,5,13]) established that if the interventions can involve many variables, only a few of them suffice to learn the causal graph.
In this paper, we investigate the problem when the size of each intervention (i.e. the number of variables that are forced to be specific values) is limited. This is a natural constraint since in practice only a few of the variables can be set by the person doing the experiment (e.g. by deactivating a gene or enforcing a specific diet). Simultaneously deactivating several genes and fixing other variables to desired values make the experiment significantly harder.

[Reviewer 6]

1) The chordality of the undirected graphs we are working on is not an assumption. Previous results [2] have shown that it is without loss of generality to assume that the unknown graph is chordal. (the directions of all the other edges can be obtained using Meek rules and removed). We thank the reviewer for raising this important comment - we will clarify it in the paper. Complete graphs are in a sense the hardest (i.e. need more interventions) of the chordal graphs and that is why we study them separately.

2) To minimize the total number of interventions the best choice for k is n/2. This unfortunately requires each intervention to involve half of the variables of the problem and may be hard to implement. We assume that k is given and our results work for all values of k. In practice, we expect interventions on few variables. So k would typically be sub-linear in n.

3) Any general graph we work with gives rise to a set of chordal graphs (see eqn (1)) after applying CI-based learning algorithms. Hence, chordal graphs are the most general graphs we encounter in this problem. Thus the statement holds true.